# Acute Kidney Injury and Septic Shock—Defined by Updated Sepsis-3 Criteria in Critically Ill Patients

**DOI:** 10.3390/jcm8101731

**Published:** 2019-10-18

**Authors:** Vin-Cent Wu, Shih-Chieh Jeff Chueh, Jui-Ting Chang, Bang-Gee Hsu, Marlies Ostermann, Tzong-Shinn Chu

**Affiliations:** 1Division of Nephrology, National Taiwan University Hospital, No. 7 Chung-Shan South Road, Zhong-Zheng District, Taipei 100, Taiwan; q91421028@ntu.edu.tw; 2Glickman Urological and Kidney Institute, Cleveland Clinic Lerner College of Medicine, Cleveland Clinic, Cleveland, 9500 Euclid Ave., Cleveland, OH 44195, USA; jeffchueh@gmail.com; 3Division of Nephrology, Department of Internal Medicine, Shin Kong Memorial Wu Ho-Su Hospital, Taipei 111, Taiwan; skhnephropgy@gmail.com; 4Division of Nephrology, Hualien Tzu Chi Hospital, Buddhist Tzu Chi Medical Foundation; and School of Medicine, Tzu Chi University, Hualien 970, Taiwan; gee.lily@msa.hinet.net; 5Department of Critical Care and Nephrology, King’s College London, Guy’s and St Thomas Hospital, London SE1 7EH, UK; 6NSARF group (National Taiwan University Hospital Study Group of ARF), Zhong-Zheng District, Taipei 100, Taiwan; 7CAKS group (Taiwan Consortium for Acute Kidney Injury and Renal Diseases), Guishan District, Taoyuan City 333, Taiwan

**Keywords:** Sepsis-3, acute dialysis, qSOFA, acute kidney injury

## Abstract

Sepsis is commonly associated with acute kidney injury (AKI), particularly in those requiring dialysis (AKI-D). To date, Sepsis-3 criteria have not been applied to AKI-D patients. We investigated sepsis prevalence defined by Sepsis-3 criteria and evaluated the outcomes of septic-associated AKI-D among critically ill patients. Using the data collected from a prospective multi-center observational study, we applied the Sepsis-3 criteria to critically ill AKI-D patients treated in intensive care units (ICUs) in 30 hospitals between September 2014 and December 2015. We described the prevalence, outcomes, and characteristics of sepsis as defined by the screening Sepsis-3 criteria among AKI-D patients, and compared the outcomes of AKI-D patients with or without sepsis using the Sepsis-3 criteria. A total of 1078 patients (median 70 years; 673 (62.4%) men) with AKI-D were analyzed. The main etiology of AKI was sepsis (71.43%) and the most frequent indication for acute dialysis was oliguria (64.4%). A total of 577 (53.3% of 1078 patients) met the Sepsis-3 criteria, and 206 among the 577 patients (19.1%) had septic shock. Having sepsis and septic shock were independently associated with 90-day mortality among these ICU AKI-D patients (hazard ratio (HR) 1.23 (*p* = 0.027) and 1.39 (*p* = 0.004), respectively). Taking mortality as a competing risk factor, AKI-D patients with septic shock had a significantly reduced chance of weaning from dialysis at 90 days than those without sepsis (HR 0.65, *p* = 0.026). The combination of the Sepsis-3 criteria with the AKI risk score led to better performance in forecasting 90-day mortality. Sepsis affects more than 50% of ICU AKI patients requiring dialysis, and one-fifth of these patients had septic shock. In AKI-D patients, coexistent with or induced by sepsis (as screened by the Sepsis-3 criteria), there is a significantly higher mortality and reduced chance of recovering sufficient renal function, when compared to those without sepsis.

## 1. Introduction

Sepsis is considered the most frequent cause of acute kidney injury (AKI) in critically ill patients in the intensive care unit (ICU) [1]. It is a heterogeneous syndrome caused by an unbalanced host response to an infection, often resulting in variable clinical signs and symptoms. Until the early 1990s, sepsis was not formally defined, and numerous different criteria were used in research and clinical practice. In 2016, the ‘Sepsis-3’ consensus definition was published. Accordingly, sepsis constitutes life-threatening organ dysfunction that is caused by a dysregulated host response to infection and defined by an acute change in total Sequential Organ Failure Assessment (SOFA) score by ≥2 points (delta SOFA). In addition, the concept of the quick SOFA (qSOFA) was introduced as a possible tool to identify patients with sepsis outside the ICU [2]. The qSOFA describes the presence of any two of the following three factors: Respiratory rate ≥ 22/min, altered mentation or systolic blood pressure ≤ 100 mmHg. Septic shock is recognized as a subset of sepsis with profound circulatory, cellular, and metabolic abnormalities as evidenced by a serum lactate concentration >2 mmol/L and vasopressor requirement to maintain a mean arterial pressure (MAP) of at least 65 mmHg in the absence of hypovolemia. Of note, the terms systemic inflammatory response syndrome (SIRS) and severe sepsis were removed.

An analysis of data from a large cohort of patients admitted to 409 hospitals in the USA in 2004–2009 revealed that more than 40% of patients with sepsis, as defined by the Sepsis-3 criteria, also had AKI [3]. In patients with AKI, mortality and long-term outcomes are worst in those treated with renal replacement (RRT), also known as ‘AKI with dialysis’ (AKI-D) [2,4]. To date, the prevalence and outcomes of sepsis defined by the Sepsis-3 criteria among AKI-D patients has not been well reported. It is unclear whether the two criteria of the Sepsis-3 definition are equally predictive when used in association with AKI-D patients.

The aims of our project were: (i) To describe the incidence, outcomes, and characteristics of sepsis, as defined by the screening Sepsis-3 criteria, among AKI-D patients; and (ii) to show the outcomes of AKI-D patients without or with sepsis.

## 2. Methods

### 2.1. Study Population

We analyzed data of the Taiwan Consortium for Acute Kidney Injury and Renal Diseases (CAKs) study. The CAKS study was approved by the institutional review boards of the participating institutions. The need for informed consent was waived because all personal data was fully de-identified and only data that were routinely collected for clinical purposes were analyzed (approval number NRPB2014050014). The CAKs study has been extensively described previously [5,6]. In brief, it is a prospective observational study of ICU patients with AKI-D admitted to one of 30 hospitals in Taiwan. The hospitals are distributed widely through the various geographical regions of Taiwan (north, middle, south, and east) and there is a 1:1 ratio of tertiary medical centers to regional hospitals in each region. We analyzed patients who had been enrolled in four distinct months (October 2014, January 2015, April 2015 and July 2015) and were followed-up for at least three months after hospital discharge. Patients receiving chronic dialysis before the index hospitalization were excluded.

### 2.2. Dialysis Initiation

The pre-determined indications for RRT protocol or algorithm or simply clinician judgement were: (1) Presence of azotemia (blood urea nitrogen (BUN) > 80 mg/dL and serum creatinine (sCr) > 2 mg/dL) and uremic symptoms (encephalopathy, pericarditis or pleurisy); (2) oliguria (urine output < 400 mL/24 h) or anuria refractory to fluid challenges and diuretics; (3) fluid overload refractory to diuretics with a central venous pressure (CVP) > 12 mmHg or pulmonary edema with a PaO2/FiO2 < 300 mmHg; (4) hyperkalemia (serum potassium > 5.5 mmol/L) refractory to medical treatment, and/or (5) metabolic acidosis (arterial pH < 7.2) [7,8,9].

### 2.3. Infection and Sepsis

The medical records of all patients were independently reviewed by two investigators to identify AKI-D patients who met the Sepsis-3 criteria at initiation of RRT. In case of any discordance, a third investigator (VCW) acted as an adjudicator.

To be classified as having sepsis, patients with a suspected or confirmed infection had to have at least two qSOFA criteria [10] or an acute increase in total SOFA score by ≥2 within the 24 h period before acute RRT was started (distribution as Appendix A). Infection was defined as body fluids with positive culture or antibiotics started as a criterion of suspected infection. In patients with a nosocomial infection in whom a pre-admission SOFA score was not available, the first SOFA value at hospital admission qualified as baseline score (1). Patients with hepatic dysfunction who received acute RRT within 24 h of ICU admission, were assigned a baseline SOFA score of four, and in case of chronic respiratory impairment, a baseline score of two was assigned [11]. In all other cases, the baseline SOFA score was considered to be zero [2]. Consistent with previous studies, missing values were considered to be normal [10].

### 2.4. Outcomes

The primary study outcome was 90-day mortality after hospital discharge. Secondary outcomes were inability to wean from acute RRT and/or a composite outcome of mortality or RRT dependence at 90 days after hospital discharge [12].

### 2.5. Clinical Data Collection

In patients with AKI-D at initiation of RRT, we extracted the following data: Baseline characteristics and demographics, severity of illness scores including SOFA score, acute physiology and chronic health evaluation II (APACHE II) score and multiple organ dysfunction syndrome (MODS) score, comorbidities and the presumed etiology of AKI. AKI was defined by the serum creatinine criteria of the Kidney Disease Improving Global Outcome classification [13]. We also calculated the AKI risk prediction score as proposed by Demirjian et al. [14] (Appendix A). The worst physiological and biochemical values during the initial 24-h period before RRT were recorded, together with the severity of illness scores and vasopressor administration at initiation of RRT [15].

The pre-determined indications for RRT were mentioned above and as in supplemental methods.

### 2.6. Statistical Analyses

Continuous data were expressed as mean ± standard deviation (SD) and group comparisons were conducted using χ 2 tests for equal proportions, t tests for normally distributed data, and Wilcoxon rank sum tests otherwise.

We performed multivariable analyses of all factors that were significant in univariate analyses, including age, sex, baseline comorbidities, indication for RRT, etiology of AKI, renal parameters and SOFA score at initiation of RRT and modality of RRT. The significance levels for entry (SLE) and for stay (SLS) were set at a conservative level of 0.15. The best candidate final logistic regression model was identified manually by dropping the covariates with a *p* value > 0.05 one at a time until all regression coefficients were significantly different from zero.

Net reclassification improvement (NRI) and integrated discrimination improvement (IDI) analysis were used to examine the role of the Sepsis-3 criteria to stratify individuals into higher or lower risk categories (reclassification) [16,17]. Focusing on 90-day mortality, an increase in NRI was calculated in a model containing the Sepsis-3 criteria in combination with the AKI risk prediction or SOFA score. The results were compared with the individual criteria of the Sepsis-3 definition. We distinguished between three risk categories (0%–40%, 40%–80%, and >80%) and reclassified the patients who died from all-causes or were still dialysis dependent at 90 days after hospital discharge (according to decision curve analysis, Appendix A).

All analyses were performed with R software, version 3.2.2 (Free Software Foundation, Inc., Boston, MA, USA), MedCalc Statistical Software, version 15.11.3 (MedCalc Software bvba, Ostend, Belgium; https://www.medcalc.org; 2015) and Stata version 12 (StataCorp LP, College Station, TX, USA) for competing-risk analysis. A *p*-value < 0.05 was considered significant.

## 3. Results

### 3.1. Patient Cohort

We enrolled a total of 1078 critically ill patients with AKI-D (median age 70 years; 673 (62.4%) male). At initiation of RRT, their median SOFA score was 12, APACHE II score 24 and MODS score was 11. A total of 577 (53.5%) patients had sepsis, of whom 206 (19.1%) met the criteria for septic shock (Figure 1). Appendix A shows the distribution of the qSOFA and SOFA scores. The main source of infection was the respiratory tract (53.6%), followed by the genitourinary tract (31.4%). The missing data were mainly come from bilirubin (*n* = 4, 3.7%) and coagulation INR (5, 4.6%).

Based on judgment by the clinicians treating the patients, the main etiologies of AKI were sepsis (71.4%), shock (57.1%) and rhabdomyolysis (14.4%). The most frequent indication for acute RRT was oliguria (64.4%), followed by fluid overload (56.0%), azotemia (54.4%) and metabolic acidosis (49.6%).

### 3.2. Impact of Sepsis

Patients with AKI-D who had sepsis (53.5%) or septic shock (19.1%) at initiation of RRT were significantly older, had better baseline renal function and a higher serum lactate result on admission to ICU compared to AKI-D patients without sepsis (Table 1).

### 3.3. Comparison between 90-Day Survivors and Non-Survivors

Mortality and composite outcome (mortality or RRT dependence) at 90 days were 62.3%, and 76.4% in the 1078 AKI-D patients, respectively (Table 2). Sepsis and septic shock were more common among non-survivors compared to survivors (63.2% and 25.6% versus 37.4% and 8.4%, respectively). Other significant differences between 90-day survivors and non-survivors were older age, a higher comorbidity score, lower baseline serum creatinine and a higher prevalence of liver cirrhosis and cancer in non-survivors.

### 3.4. Sepsis-3 Criteria versus 90-Day Outcomes

Multivariable analysis showed that AKI-D patients with sepsis or septic shock at initiation of RRT had a significantly higher risk of mortality at 90 days compared to AKI-D patients without sepsis (Table 3 and Figure 2a). There was a positive correlation between the Sepsis-3 criteria and APACHE II score (*r* = 0.385, *p* < 0.001), SOFA score (*r* = 0.391, *p* < 0.001) and AKI risk prediction score (*r* = 0.359, *p* < 0.001). AKI-D patients with septic shock had a greater incremental increase in 90-day mortality across all deciles of APACHE II at initiation of RRT compared to AKI-D patients with qSOFA ≥ 2 (Appendix A).

### 3.5. Sepsis per Sepsis-3 Criteria versus the Risk of Dialysis Dependence

Multivariable analysis showed that after controlling for mortality, AKI-D patients with septic shock who survived had a significantly lower likelihood of weaning from dialysis, when compared to AKI-D patients without sepsis (hazard ratio (HR), 0.65, *p* = 0.026) (Table 3). There was no significant difference in likelihood of weaning from dialysis at 90 days between AKI-D patients without sepsis and AKI-D patients with sepsis but without septic shock (HR 0.96, *p* = 0.760) (Table 3, Figure 2b).

### 3.6. Evaluation of Sepsis-3 Criteria in Combination with AKI Risk Prediction Score and SOFA Score versus the 90 Days Mortality

Combining the Sepsis-3 criteria with the AKI risk prediction score at initiation of RRT led to a significant increase in risk stratification (total NRI = 0.07; 95% CI, 0.02–0.11; *p* < 0.01). The majority of this effect came from those without death (NRI event = 0.04; 95% CI, 0.01–0.07; *p* = 0.039). Likewise, the total IDI was significant (0.02, 95% CI, 0.01–0.02; *p* < 0.001).

In case of sequential diagnosis of sepsis according to the Sepsis-3 criteria, we added Sepsis-3 to the qSOFA criteria in estimating the risk of 90-day mortality after initiation of dialysis. This led to a significant increase in risk stratification (total NRI = 0.11; 95% CI, 0.04–0.19; *p* = 0.004). The majority of this effect came from those without death (NRI event = 0.07; *p* = 0.03), whereas the NRI with death was 0.05 (NRI non-event = 0.04, *p* = 0.049) (Appendix A). Similarly, the total integrated discrimination improvement (IDI) was significant at 0.06 (95% CI, 0.03 to 0.05; *p* < 0.001).

## 4. Discussion

To our best knowledge, this is the first study that applied the most recent Sepsis-3 screening criteria to the patients with AKI-D. The key findings of this large multi-center prospective study are that sepsis affects 53.5% of patients with AKI-D, and that at time of RRT 19.1% of patients had septic shock. The chances of renal recovery at 90 days were significantly lower in AKI-D patients with septic shock compared to those without sepsis. Presence of sepsis per Sepsis-3 criteria in AKI-D patients is associated with higher mortality rate and composite outcome at 90 days. Combining the Sepsis-3 criteria with the AKI risk score or SOFA criteria led to a further improvement in risk identification.

A 53.5% prevalence of sepsis among AKI-D patients is slightly higher than previously reported [18]. This may be a reflection of our specific patient cohort of critically ill ICU patients or a result of using the Sepsis-3 criteria. Similarly, in-hospital mortality rate of this cohort was high, including its non-septic controls (44.1%), which again may be explained by the characteristics of our patient population (older age, high comorbidity and acute severity of illness scores) and the criteria used to identify patients.

### 4.1. Association of Mortality and Non-Recovery from Dialysis

Patients with AKI requiring RRT constitute a high-risk group. An accurate prognostic assessment is crucial for clinical management and planning of future care. The Sepsis-3 criteria identified AKI-D patients with a suspected or confirmed infection who were at increased risk of mortality, and a combo endpoint of mortality or dialysis dependence at 90 days. Our data also showed that the criteria to define septic shock (i.e., a raised serum lactate level and the need for inotropic support) indeed identified the subgroup of patients with the highest mortality (>80%).

The Sepsis-3 criteria correlated with SOFA and APACHE II scores at initiation of dialysis which underpins the potential use of Sepsis-3 in a critical care setting. It can be hypothesized that a higher Sepsis-3 score also reflects a higher degree of systemic inflammation.

Data on the risk of long-term dialysis dependence in AKI-D patients with sepsis are conflicting [18,19], similar to a French multicentric study [19]. We found a statistically significant trend towards reduced likelihood of recovery from dialysis in AKI-D patients with septic shock compared to those without sepsis. In contrast, Bagshaw and colleagues analyzed data from 2000/2001 and reported improved renal recovery in patients with septic AKI compared to patients without sepsis [18]. It is important to note that there were differences in patient characteristics, criteria to define sepsis and clinical care. Moreover, our analysis has extra strength by including mortality as a competing outcome for analysis of dialysis dependence.

### 4.2. Sepsis-3 Criteria and Outcome

It is important to emphasize that most AKI patients already had a SOFA score of two or more at the time when RRT was initiated simply as a result of AKI and oliguria. It is possible that different delta SOFA cut-off points are necessary for this patient cohort to differentiate sepsis from non-sepsis (Appendix A). As such, our results provide confirmation that AKI-D patients with sepsis, as defined by Sepsis-3 criteria, had higher mortality and less withdraw from dialysis in AKI-D patients with septic shock.

Combining the Sepsis-3 criteria with a clinical AKI risk prediction score resulted in greater IDI and NRI, and improved the ability associating subsequent death. Given the lack of appropriate risk stratification tools for septic AKI patients requiring RRT, the new Sepsis-3 criteria may associate patients outcome (Appendix A).

### 4.3. Strengths and Limitations

This is the first study that applied the new consensus criteria for sepsis to AKI-D patients, a cohort of patients that is known to have a high risk of sepsis and also a high risk of poor outcomes. Using a large multi-center national database with prospectively collected representative data from 30 ICUs, we showed for the first time that the Sepsis-3 criteria identified a group of patients that were at higher risk of dying or remaining dialysis dependent after discharge. With complete follow-up for 90 days after discharge from hospital, we focused on patient-centered outcomes (mortality and long-term dialysis dependence) and provide important data for a group of patients that is commonly seen in ICUs worldwide. Further in-depth studies are mandatory before we can make any positive comment on this issue.

The limitations of our study are related to any observational cohort study and include the possibility of both unmeasured and residual confounding factors. We could not identify how many patients received early goal-directed therapy in our cohort, however the nationwide education program instituted in Taiwan is able to positively change critical-care physician behavior in sepsis care following the Surviving Sepsis Campaign guidelines. We also acknowledge that we recorded the worst value of SOFA criteria within 24 h before initiation of dialysis. Although this approach is consistent with clinical practice, the daily SOFA score or qSOFA value may not reflect the value immediately before initiation of RRT.

## 5. Conclusions

More than half of the critically ill AKI patients treated with dialysis had sepsis, as defined by the Sepsis-3 criteria, at dialysis initiation, and one-fifth of AKI-D patients had septic shock screened by Sepsis-3 criteria. Having sepsis and septic shock were independently associated with 90-day mortality among these ICU AKI-D patients. Among survivors, AKI-D patients with septic shock had a significantly reduced chance of recovering sufficient renal function to wean-off dialysis, when compared to those without sepsis. These findings provide support for the use of Sepsis-3 criteria in the AKI-D patients.

## Figures and Tables

**Figure 1 jcm-08-01731-f001:**
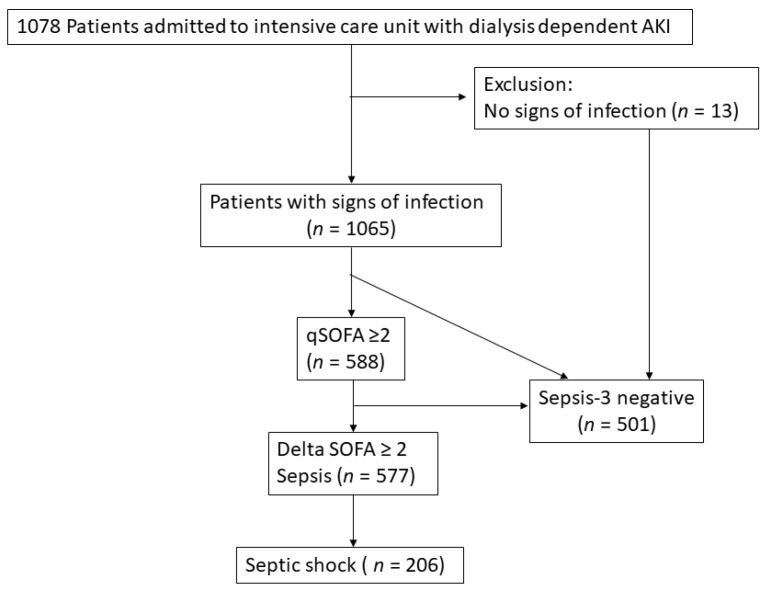
Algorithm of enrollee.

**Figure 2 jcm-08-01731-f002:**
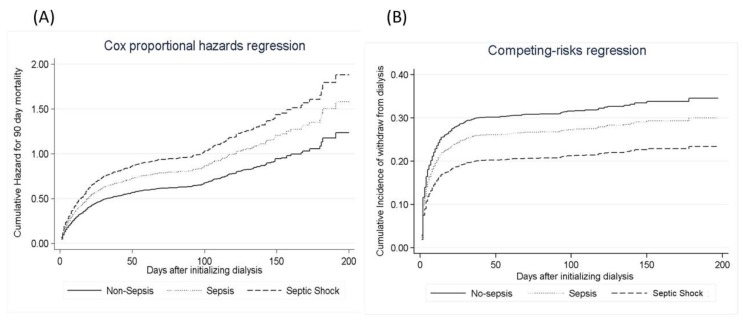
Cox proportional hazards models. (**A**) Cox proportional hazards models are plotted to model the probability of free from 90 days mortality, stratified by Sepsis-3 status. (**B**) Model the risk of chronic dialysis, taking mortality as a competing risk.

**Table 1 jcm-08-01731-t001:** Clinical characteristics of patients with and without sepsis.

	Non-Sepsis	Sepsis	Septic Shock	*p* Value
	(*n* = 501)	(*n* = 371)	(*n* = 206)	
Patient characteristics				
Age, median (range)	71.8 (60.6–80.3)	69.3 (57.6–79.7)	65.8 (54.3–76.3)	0.011
Male gender, *n* (%)	299 (59.68%)	228 (61.46%)	146 (70.87%)	0.018
BMI, median (range)	23.9 (21.4–27.2)	24 (21–27.2)	23.9 (21–26.9)	0.579
Charlson comorbidity index	7 (5–9)	7 (5–9)	6 (5–8)	0.020
Baseline sCr (mg/dL), median (range)	1.7 (1–3.3)	1.4 (0.9–2.4)	1 (0.8–1.7)	<0.001
eGFR (ml/min/1.73 m^2^), median (range)	32.5 (15.5–63.3)	44 (21.9–73.8)	63.9 (35.2–88)	<0.001
Comorbidities, *n* (%)				
Diabetes mellitus, *n* (%)	276 (55.09%)	189 (50.94%)	97 (47.09%)	0.131
Liver cirrhosis, *n* (%)	56 (11.18%)	58 (15.63%)	40 (19.42%)	0.011
COPD, *n* (%)	43 (8.58%)	29 (7.82%)	12 (5.83%)	0.462
CAD, *n* (%)	156 (31.14%)	102 (27.49%)	52 (25.24%)	0.233
CVA, *n* (%)	78 (15.57%)	59 (15.90%)	22 (10.68%)	0.185
Hemiplegia, *n* (%)	22 (4.39%)	20 (5.39%)	6 (2.91%)	0.383
GI bleeding, *n* (%)	129 (25.75%)	109 (29.38%)	58 (28.16%)	0.479
Dementia, *n* (%)	11 (2.20%)	12 (3.23%)	9 (4.37%)	0.282
Cancer, *n* (%)	86 (17.17%)	79 (21.29%)	63 (30.58%)	<0.001
Congestive heart failure, *n* (%)	194 (48.7%)	159 (58.49%)	68 (44.66%)	<0.001
Laboratory data at ICU admission				
BUN (mg/dL), median (range)	61 (34.5–95.9)	57.7 (27–92)	42 (23.2–68.8)	<0.001
sCr (mg/dL), median (range)	3.7 (2–6.4)	2.7 (1.4–5)	2.3 (1.3–3.8)	<0.001
Lactate (mmol/L), median (range)	2.5 (1.4–5.6)	2.4 (1.3–4.8)	6.3 (2.9–10)	<0.001
Etiology of AKI (except sepsis), *n* (%)				
Shock	225 (44.91%)	203 (54.72%)	188 (91.26%)	<0.001
Cardiorenal syndrome	206 (41.12%)	134 (36.12%)	53 (25.73%)	<0.001
Drug nephrotoxicity	26 (5.19%)	18 (4.85%)	10 (4.85%)	0.969
Rhabdomyolysis	34 (6.79%)	24 (6.47%)	23 (11.17%)	0.086
Intravascular hemolysis	16 (3.19%)	10 (2.70%)	8 (3.88%)	0.735
Hepatorenal	26 (5.19%)	22 (5.93%)	21 (10.19%)	0.043
ATIN	4 (0.80%)	5 (1.35%)	0 (0%)	0.276
Contrast	38 (7.58%)	24 (6.47%)	13 (6.31%)	0.750
Obstructive	8 (1.60%)	3 (0.81%)	1 (0.49%)	0.472
Others	117 (23.35%)	56 (15.09%)	24 (11.65%)	<0.001
At initiation of RRT				
Urine output (mL/24 h), median (range)	450 (150–1095)	250 (70–620)	130 (50–418)	<0.001
AKI risk prediction score, median (range)	22 (17–28)	27 (21–33)	33.5 (26–40)	<0.001
Lactate (mmol/L), median (range)	2.2 (1.3–5.2)	1.6 (1–3.1)	6.6 (3.4–10.7)	<0.001
SOFA score, median (range)	10 (7–13)	12 (10–15)	15 (13–17)	<0.001
qSOFA, median (range)	1 (1–1)	2 (2-3)	2 (2–3)	<0.001
APACHE II score, median (range)	20 (16–25)	25 (21–30)	27 (22.8–33)	<0.001
MODS score, median (range)	9 (7–11)	12 (10–14)	12 (10–15)	<0.001
Site of infection, *n* (%)				
Respiratory	227 (45.31%)	237 (63.88%)	114 (55.34%)	<0.001
GU	156 (31.14%)	129 (34.77%)	53 (25.73%)	0.080
Bacteremia	96 (19.16%)	84 (22.64%)	57 (27.67%)	0.043
Abdomen	41 (8.18%)	40 (10.78%)	33 (16.02%)	0.009
Others	56 (11.18%)	38 (10.24%)	21 (10.19%)	0.880
Indication for dialysis, *n* (%)				
Azotemia	291 (58.08%)	225 (60.65%)	70 (33.98%)	<0.001
Fluid overload	245 (48.90%)	225 (60.65%)	134 (65.05%)	<0.001
Electrolyte imbalance	190 (37.92%)	148 (39.89%)	79 (38.35%)	0.835
Metabolic acidosis	210 (41.92%)	192 (51.75%)	133 (64.56%)	<0.001
Oliguria	275 (54.89%)	255 (68.73%)	164 (79.61%)	<0.001
Uremic encephalopathy	46 (9.18%)	26 (7.01%)	6 (2.91%)	0.014
Dialysis modality, *n* (%)				<0.001
CVVH	128 (25.55%)	110 (29.65%)	133 (64.56%)	
IHD	334 (66.67%)	252 (67.92%)	58 (28.16%)	
SLEDD	39 (7.78%)	9 (2.43%)	15 (7.28%)	
Outcomes of interest				
Dialysis days in hospital, median (range)	12 (4–26)	10 (4–27)	6 (3–15)	0.012
Hospital mortality, *n* (%)	221 (44.11%)	228 (61.46%)	167 (81.07%)	<0.001
90-day ICU free days	63 (0–85)	1 (0–78)	0 (0–0)	<0.001
90-day hospital free days	30 (0–70)	0 (0–53)	0 (0–0)	<0.001
90-day mortality, *n* (%)	246 (49.10%)	253 (68.19%)	172 (83.49%)	<0.001
90-day composite outcome, *n* (%)	352 (70.26%)	296 (79.78%)	175 (84.95%)	<0.001

Paired comparison between the groups. Abbreviations: AKI, acute kidney injury; APACHE; acute physiology and chronic health evaluation; ATIN, acute tubule-interstitial nephritis; BMI, body mass index; BUN, blood urea nitrogen; CAD, coronary artery disease; CABG, coronary artery bypass graft; COPD, chronic obstructive pulmonary disease; CPB, cardiopulmonary bypass; CVA, cerebrovascular accident; CVVH, continuous veno-venous hemofiltration; CPR, cardio-pulmonary resuscitation; eGFR, estimated glomerular filtration rate; GSC, Glasgow Coma Scale; GI, gastrointestinal; GU, genitourinary; IABP, intra-aortic balloon pump; ICU, intensive care unit; IHD, intermittent hemodialysis; IQR, interquartile range; MODS, multiple organ dysfunction; RRT, renal replacement therapy; sCr, serum creatinine; SLEDD, sustained low efficiency daily dialysis; SOFA, Sequential Organ Failure Assessment.

**Table 2 jcm-08-01731-t002:** Clinical characteristics of survivors and non-survivors.

	All	90 Day Survivors	90 Day Mortality	*p* Value	No Dialysis Dependence or Mortality at 90 Days	Dialysis Dependence or Mortality at 90 Days	*p* Value
	(*n* = 1078)	(*n* = 406)	(*n* = 672)		(*n* = 254)	(*n* = 824)	
Baseline characteristics							
Age, median (range)	70 (57.8–79.5)	69 (56.7–77.4)	71 (58.9–81)	0.014	67.7 (53.9–76.8)	70.9 (59.8–80.4)	<0.001
Male gender, *n* (%)	673 (62.43%)	247 (60.84%)	426 (63.39%)	0.401	158 (62.20%)	515 (62.50%)	0.932
BMI, median (range)	23.95 (21.2–27.1)	24.2 (21.5–27.6)	23.7 (21–26.8)	0.870	24.6 (22–27.9)	23.7 (21–26.8)	0.598
Charlson comorbidity index, median (range)	7 (5–9)	7 (5–8.3)	7 (5–9)	0.001	6 (4–8)	7 (5–9)	<0.001
Baseline sCr (mg/dL), median (range)	1.4 (0.9–2.7)	1.8 (1–3.6)	1.3 (0.9–2.3)	<0.001	1.2 (0.9–2.1)	1.5 (0.9–2.8)	<0.001
eGFR (mL/min/1.73 m^2^), median (range)	41.79 (20.4–73.6)	31.9 (14.6-64.3)	48.2 (24.4-77.1)	<0.001	49.9 (25.3-78)	40.5 (18.2-71.7)	0.040
Comorbidities							
Diabetes mellitus, *n* (%)	562 (52.13%)	228 (56.16%)	334 (49.70%)	0.040	139 (54.72%)	423 (51.33%)	0.344
Liver cirrhosis, *n* (%)	154 (14.29%)	26 (6.40%)	128 (19.05%)	<0.001	19 (7.48%)	135 (16.38%)	<0.001
COPD, *n* (%)	84 (7.79%)	31 (7.64%)	53 (7.89%)	0.881	19 (7.48%)	65 (7.89%)	0.832
CAD, *n* (%)	310 (28.76%)	130 (32.02%)	180 (26.79%)	0.066	77 (30.31%)	233 (28.28%)	0.530
CVA, *n* (%)	159 (14.75%)	61 (15.02%)	98 (14.58%)	0.843	30 (11.81%)	129 (15.66%)	0.131
Hemiplegia, *n* (%)	48 (4.45%)	19 (4.68%)	29 (4.32%)	0.779	11 (4.33%)	37 (4.49%)	0.914
GI bleeding, *n* (%)	296 (27.46%)	89 (21.92%)	207 (30.80%)	0.002	53 (20.87%)	243 (29.49%)	0.007
Dementia, *n* (%)	32 (2.97%)	7 (1.72%)	25 (3.72%)	0.061	6 (2.36%)	26 (3.16%)	0.515
Cancer, *n* (%)	228 (21.15%)	60 (14.78%)	168 (25.00%)	<0.001	40 (15.75%)	188 (22.82%)	0.016
Congestive heart failure, *n* (%)	553 (51.30%)	205 (50.49%)	348 (50.79%)	0.787	122 (48.41%)	431 (52.31%)	0.019
Parameters at ICU admission							
BUN (mg/dL), median (range)	56 (29.2–89)	63 (35–91)	50.5 (26.4–88)	0.001	51 (26–80.5)	57.7 (30–91.6)	0.192
sCr (mg/dL), median (range)	3 (1.7–5.5)	4.1 (2.2–6.9)	2.6 (1.4–4.5)	<0.001	3 (1.9–5.3)	3 (1.6–5.6)	0.984
Lactate (mmol/L), median (range)	3.1 (1.7–7)	2.6 (1.3–5.2)	3.7 (2–8.6)	<0.001	3 (1.6–6.1)	3.2 (1.7–7.3)	0.030
Etiology of AKI, *n* (%)							
Shock, *n* (%)	616 (57.14%)	165 (40.64%)	451 (67.11%)	<0.001	132 (51.97%)	484 (58.74%)	0.057
Sepsis, *n* (%)	770 (71.43%)	242 (59.61%)	528 (78.57%)	<0.001	153 (60.24%)	617 (74.88%)	<0.001
Cardiorenal syndrome, *n* (%)	393 (36.46%)	170 (41.87%)	223 (33.18%)	0.010	93 (36.61%)	300 (36.41%)	0.952
Nephrotoxic drugs, *n* (%)	54 (5.01%)	27 (6.65%)	27 (4.02%)	0.055	22 (8.66%)	32 (3.88%)	0.002
Rhabdomyolysis, *n* (%)	81 (7.51%)	33 (8.13%)	48 (7.14%)	0.552	28 (11.02%)	53 (6.43%)	0.015
Intravascular hemolysis, *n* (%)	34 (3.15%)	14 (3.45%)	20 (2.98%)	0.667	11 (4.33%)	23 (2.79%)	0.220
Hepatorenal syndrome, *n* (%)	69 (6.40%)	4 (0.99%)	65 (9.67%)	<0.001	4 (1.57%)	65 (7.89%)	<0.001
ATIN, *n* (%)	9 (0.83%)	5 (1.23%)	4 (0.60%)	0.309	2 (0.79%)	7 (0.85%)	0.999
Contrast exposure, *n* (%)	75 (6.96%)	33 (8.13%)	42 (6.25%)	0.240	22 (8.66%)	53 (6.43%)	0.222
Obstruction, *n* (%)	12 (1.11%)	6 (1.48%)	6 (0.89%)	0.375	3 (1.18%)	9 (1.09%)	1.000
Others, *n* (%)	197 (18.27%)	103 (25.37%)	94 (13.99%)	<0.001	61 (24.02%)	136 (16.50%)	0.007
Parameters at RRT initiation							
Urine output (mL/24 h), median (range)	300 (90–822.5)	490 (160–1223)	204 (70–595)	<0.001	520 (180–1305)	250 (75–670)	<0.001
AKI risk prediction score	25 (19–33)	21 (16–28.3)	27.5 (22–35)	<0.001	22 (17–29.3)	26 (21–34)	<0.001
Lactate (mmol/L), median (range)	3.2 (1.6–7.6)	2.3 (1.2–5.4)	3.9 (1.9–9.1)	<0.001	2.8 (1.4–6.5)	3.3 (1.6–8.2)	0.090
SOFA score, median (range)	12 (8–15)	9 (7–12)	13 (10–16)	<0.001	10 (7–13)	12 (9–15)	<0.001
qSOFA, median (range)	2 (1–2)	1 (1–2)	2 (1–2)	<0.001	1 (1–2)	2 (1–2)	<0.001
qSOFA ≥ 2, *n* (%)	582 (53.99%)	153 (37.66%)	429 (63.84%)	<0.001	107 (42.13%)	475 (57.65%)	<0.001
APACHE II score, median (range)	24 (19–28)	20 (16–25)	25 (21–30)	<0.001	21 (16–26)	24 (20–29)	<0.001
MODS score, median (range)	11 (9–13)	10 (7–12)	11 (9–14)	<0.001	10 (7.5–13)	11 (9–13)	0.008
Sepsis 3 criteria							
Sepsis, *n* (%)	577 (53.53%)	152 (37.44%)	425 (63.24%)	<0.001	106 (41.73%)	471 (57.16%)	<0.001
Septic shock, *n* (%)	206 (19.11%)	34 (8.37%)	172 (25.60%)	<0.001	31 (12.20%)	175 (21.24%)	0.001
Site of infection, *n* (%)							
Respiratory	578 (53.62%)	188 (46.31%)	390 (58.04%)	<0.001	1113 (44.49%)	465 (56.43%)	0.001
GU	338 (31.35%)	134 (33.00%)	204 (30.36%)	0.364	83 (32.68%)	255 (30.95%)	0.603
Bacteremia	237 (21.99%)	60 (14.78%)	177 (26.34%)	<0.001	43 (16.93%)	194 (23.54%)	0.026
Abdomen	114 (10.58%)	39 (9.61%)	75 (11.16%)	0.421	32 (12.60%)	82 (9.95%)	0.230
Others	115 (10.67%)	37 (9.11%)	78 (11.61%)	0.199	25 (9.84%)	90 (10.92%)	0.626
Indication for RRT							
Azotemia, *n* (%)	586 (54.36%)	220 (54.19%)	366 (54.46%)	0.929	113 (44.49%)	473 (57.40%)	<0.001
Fluid overload, *n* (%)	604 (56.03%)	200 (49.26%)	404 (60.12%)	0.001	132 (51.97%)	472 (57.28%)	0.136
Electrolyte imbalance, *n* (%)	417 (38.68%)	160 (39.41%)	257 (38.24%)	0.704	108 (42.52%)	309 (37.50%)	0.151
Metabolic acidosis, *n* (%)	535 (49.63%)	180 (44.33%)	355 (52.83%)	0.007	114 (44.88%)	421 (51.09%)	0.084
Oliguria, *n* (%)	694 (64.38%)	205 (50.49%)	489 (72.77%)	<0.001	122 (48.03%)	572 (69.42%)	<0.001
Uremic encephalopathy, *n* (%)	78 (7.24%)	38 (9.36%)	40 (5.95%)	0.036	16 (6.30%)	62 (7.52%)	0.510
First Dialysis modality, *n* (%)				<0.001			0.239
CVVH	371 (34.42%)	97 (23.89%)	274 (40.77%)		79 (31.10%)	292 (35.44%)	
IHD	644 (59.74%)	289 (71.18%)	355 (52.83%)		163 (64.17%)	481 (58.37%)	
SLEDD	63 (5.84%)	20 (4.93%)	43 (6.40%)		12 (4.72%)	51 (6.19%)	
Outcomes of interest							
90-day ICU free days	7 (0–81)	81 (69–86)	0(0–1)	<0.001	80 (68–86)	0 (0–54.5)	<0.001
90-day hospital free days	0 (0–59.25)	63 (44.5–76)	0(0–0)	<0.001	63 (42–76)	0 (0–2.5)	<0.001


Days of RRT in hospital, median (range)	10 (4–24)	11 (3–26)	9.5 (4–22.8)	0.461	5.5 (2–13.3)	12 (4–27.8)	<0.001

Abbreviations: AKI, acute kidney injury; APACHE; acute physiology and chronic health evaluation; ATIN, acute tubule-interstitial nephritis; BMI, body mass index; BUN, blood urea nitrogen; CAD, coronary artery disease; CABG, coronary artery bypass graft; COPD, chronic obstructive pulmonary disease; CPB, cardiopulmonary bypass; CVA, cerebrovascular accident; CVVH, continuous veno-venous hemofiltration; CPR, cardio-pulmonary resuscitation; eGFR, estimated glomerular filtration rate; GSC, Glasgow Coma Scale; GI, gastrointestinal; GU, genitourinary; IABP, intra-aortic balloon pump; ICU, intensive care unit; IHD, intermittent hemodialysis; IQR, interquartile range; MODS, multiple organ dysfunction; RRT, renal replacement therapy; sCr, serum creatinine; SLEDD, sustained low efficiency daily dialysis; SOFA, Sequential Organ Failure Assessment.

**Table 3 jcm-08-01731-t003:** Multivariable risk model for hospital mortality or composite outcome at discharge.

Sepsis	Non-Shock Sepsis vs Non-Sepsis			Septic Shock vs. Non-Sepsis		
Outcome of interests	Hazard Ratio	95% CI	*p*	Hazard Ratio	95% CI	*p*
Hospital mortality	1.12	0.91–1.37	0.276	1.48	1.17–1.88	0.001
Hospital composite outcomes	0.97	0.80–1.17	0.732	1.24	1.08–1.47	0.047
For 90-day mortality	1.23	1.02–1.47	0.027	1.39	1.11–1.75	0.004
For 90-day composite outcome	1.26	1.03–1.53	0.022	1.45	1.15–1.83	0.002
For 90-day weaning from dialysis	0.96	0.76–1.22	0.760	0.65	0.45–0.95	0.026

*p*; paired comparison between the groups. All relevant covariates included in the multi-variable analysis, including age, sex, baseline comorbidities, indication for dialysis, etiology of AKI, kidney profile and SOFA score at dialysis initiation, dialysis modality, and some of their interactions. Abbreviations: AKI, acute kidney injury; CI, confidence interval.

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
