# Peer review of "Acute Kidney Injury and Septic Shock—Defined by Updated Sepsis-3 Criteria in Critically Ill Patients"

_jcm, 2019, doi:10.3390/jcm8101731_

Round 1

Reviewer 1 Report

Well done observational study applying sepsis-3 criteria to AKI-D patients. Authors have presented the study well. It is well known that AKI-D patients with septic shock might have severe ATN with poor renal prognosis as well as higher mortality from clinical perspective  but as noted by the authors this is one of few studies which applied the use of sepsis-3 criteria published in JAMA 2016 which proves the mortality as well as renal recovery is extremely poor. 

Author Response

We are sending a revised manuscript, # jcm-599244, entitled “Acute kidney injury and septic shock– defined by updated Sepsis-3 criteria in critically ill patients” for your consideration of publication as an original article in JOURNAL OF CLINICAL MEDICINE. We have extensively revised the manuscript according to the reviewers’ comments. Particularly, we either answered by providing more information, or reproved by offering clear evidences to all those important questions raised by reviewer #3 explicitly; as detailed below.

We feel that we have made the interesting points more clearly elaborated in this revised manuscript, and hope that you find it suitable for publication. Below are the point-by-point responses to the reviewers’ comments.

Many thanks again for your consideration given to our study.

*Correspondences and reprint requests to Marlies Ostermann, MD, PhD; Department of Critical Care and Nephrology, King's College London, Guy's and St Thomas Hospital, Foundation Hospital, London, SE1 7EH, UK. Email, Marlies.Ostermann@gstt.nhs.uk

Reply to Reviewers

Reviewer 1 Comments:

Well done observational study applying sepsis-3 criteria to AKI-D patients. Authors have presented the study well. It is well known that AKI-D patients with septic shock might have severe ATN with poor renal prognosis as well as higher mortality from clinical perspective but as noted by the authors this is one of few studies which applied the use of sepsis-3 criteria published in JAMA 2016 which proves the mortality as well as renal recovery is extremely poor. 

Response. Thank you for your valuable comments.

Reviewer 2 Report

This paper tries to understand the severity of sepsis related AKI- D and outcomes related to this. The paper is well written and provides significant data and analyses to support the findings that are so stated. Although many times clinically seen, this paper nicely documents and tries to understand particular aspects of the clinical morbid outcomes of sepsis and septic shock AKI.

Author Response

We are sending a revised manuscript, # jcm-599244, entitled “Acute kidney injury and septic shock– defined by updated Sepsis-3 criteria in critically ill patients” for your consideration of publication as an original article in JOURNAL OF CLINICAL MEDICINE. We have extensively revised the manuscript according to the reviewers’ comments. Particularly, we either answered by providing more information, or reproved by offering clear evidences to all those important questions raised by reviewer #2 explicitly; as detailed below.

We feel that we have made the interesting points more clearly elaborated in this revised manuscript, and hope that you find it suitable for publication. Below are the point-by-point responses to the reviewers’ comments.

Many thanks again for your consideration given to our study.

*Correspondences and reprint requests to Marlies Ostermann, MD, PhD; Department of Critical Care and Nephrology, King's College London, Guy's and St Thomas Hospital, Foundation Hospital, London, SE1 7EH, UK. Email, Marlies.Ostermann@gstt.nhs.uk

Reviewer 2 Comments:

This paper tries to understand the severity of sepsis related AKI- D and outcomes related to this. The paper is well written and provides significant data and analyses to support the findings that are so stated. Although many times clinically seen, this paper nicely documents and tries to understand particular aspects of the clinical morbid outcomes of sepsis and septic shock AKI.

Response. Thank you for your valuable comments. We have further revised some typo and English.

Reviewer 3 Report

I read your paper with great interest. Here are some comments:

Major:

Sepsis is an Exposure for AKI-D (as Outcome), for outcome to correlate with exposure, it is necessary that exposure happens before the outcome. I fail to detect any data suggesting sepsis onset time and AKI onset time and their temporal relationship. That being said, you must describe that none of the AKI-D patients already had AKI-D when they were diagnosed with sepsis. if they did, those patients need to be excluded and re-run the analysis. Your primary aim was to establish prevalence of sepsis in AKI-D patients. but you reported survivor vs non-survivor table as table 1. Your table 2 could be your table 1 and then you can report current table 1 as post-hoc analysis afterwards. ICU LOS and Hospital LOS are shorter in non-survivors, it shows more patients died in that group and died quickly. reporting ICU free days and hospital free days will be more appropriate. sepsis mortality should not be reported without talking about EDGT status in each group. please make sincere attempts to find that data or include in your discussion section. The mortality seems to be very high given moderate APACHE II score and SOFA score.  you did adjust for baseline co-morbidities but not with APACHE II. Pelase make sure you adjust for APACHE-II. and if you are adding SOFA there, please be aware of collinearity. Having a Master statistician look at the analysis may help with some of these issues.

Author Response

We are sending a revised manuscript, # jcm-599244, entitled “Acute kidney injury and septic shock– defined by updated Sepsis-3 criteria in critically ill patients” for your consideration of publication as an original article in JOURNAL OF CLINICAL MEDICINE. We have extensively revised the manuscript according to the reviewers’ comments. Particularly, we either answered by providing more information, or reproved by offering clear evidences to all those important questions raised by reviewer #3 explicitly; as detailed below.

We feel that we have made the interesting points more clearly elaborated in this revised manuscript, and hope that you find it suitable for publication. Below are the point-by-point responses to the reviewers’ comments.

Many thanks again for your consideration given to our study.

*Correspondences and reprint requests to Marlies Ostermann, MD, PhD; Department of Critical Care and Nephrology, King's College London, Guy's and St Thomas Hospital, Foundation Hospital, London, SE1 7EH, UK. Email, Marlies.Ostermann@gstt.nhs.uk

Reviewer 3 Comments:

Comments and Suggestions for Authors

I read your paper with great interest. Here are some comments:

Major:

Sepsis is an Exposure for AKI-D (as Outcome), for outcome to correlate with exposure, it is necessary that exposure happens before the outcome. I fail to detect any data suggesting sepsis onset time and AKI onset time and their temporal relationship. That being said, you must describe that none of the AKI-D patients already had AKI-D when they were diagnosed with sepsis. if they did, those patients need to be excluded and re-run the analysis.

Your primary aim was to establish prevalence of sepsis in AKI-D patients. but you reported survivor vs non-survivor table as table 1. Your table 2 could be your table 1 and then you can report current table 1 as post-hoc analysis afterwards. ICU LOS and Hospital LOS are shorter in non-survivors, it shows more patients died in that group and died quickly. reporting ICU free days and hospital free days will be more appropriate.

sepsis mortality should not be reported without talking about EDGT status in each group. please make sincere attempts to find that data or include in your discussion section.

The mortality seems to be very high given moderate APACHE II score and SOFA score.  you did adjust for baseline co-morbidities but not with APACHE II. Pelase make sure you adjust for APACHE-II. and if you are adding SOFA there, please be aware of collinearity. Having a Master statistician look at the analysis may help with some of these issues. 

Response: Thank you for your valuable comments.

After re-examining our patients recruited in this study, we did not find a case with temporal sequence of AKI dialysis first and later diagnosis of sepsis. However, regarding the exposure-outcome relationship questioned by the reviewer, we did not emphasize the causative relationship of sepsis versus AKI-D in our article. As AKI-D is most commonly multifactorial, it might be challenging to pin-point it was solely led by sepsis. As we stated in the Abstract and the Main text that these AKI-D could be coexistent with or induced by sepsis. In the study, the primary outcome (end-point of the study) was 90-day mortality after hospital discharge. Secondary outcomes were inability to wean from acute RRT and/or a composite outcome of mortality or RRT dependence at 90 days after hospital discharge. What our data showed was that a majority (53.3%) of these critically ill ICU patients had some suspicious signs of infections at AKI requiring dialysis. We found that at the time of RRT, one-fifth of the patients had septic shock. Therefore both sepsis diagnosed before or at AKI-D were included, and septic AKI patients and non-septic AKI patients who required dialysis were included in this analysis. The non-septic AKI-D patients were considered as controls. According to your comment, we have changed the orders of table 1 and table 2. We have revised the parameters in Table 1 and 2 to ‘90 days ICU free days’ and ‘90 days hospital free days’. The care for (suspicious) septic patients in Taiwan has been standardized and followed the guideline of Early goal-directed therapy (EGDT)/and the Surviving Sepsis Campaign (SSC), as that was the basic requirement of an intnesivist/emergency specialist in Taiwan. So a quick answer to the reviewer’s question is that all the patient care were compliant to EGDT principle.. (Rhodes et al., 2017). Following the publication of the SSC in 2004 (Dellinger et al., 2004), a nationwide effort of educational program was launched by the joint Committee of Taiwan Critical Care Medicine. This committee consisted of three medical societies, namely Critical Care Medicine, Emergency and Critical Care Medicine, and Pulmonary and Critical Care Medicine (with a total number of board certified specialists = 2102 at 2013). Briefly, the education program consisted of at least 10-hours of training sessions for each intensivist. A Taiwanese intensivist was required to complete the training program to take critical care license examinations and to extend their licenses. The education program has been held at least 5 times a year since 2004 and almost all physicians who work in an ICU have completed the training course. After launching this program in Taiwan, the cumulative mortality for severe sepsis was 48.2% from 2000 to 2003 and this had decreased to 45.9% from 2005 to 2008(Chen, Chang, Pu, & Tang, 2013). From Taiwan Data, the nationwide education program instituted in Taiwan is able to positively change critical physician behavior in sepsis care following the SSC guidelines and may have some benefits in patient outcome for severe sepsis. Although we could not identify how many patients received EDGT, it is reasonable to suppose every enrollee received of EDGT management after admitted to hospital. We have condensed the statement to study limitation. (Please see page 13, the last paragraph.) The median APACHE II score is 24 in our cohort and 90 day mortality rate after hospital discharge is 62.3 %. In patients who initiated acute hemodialysis , the APCEH II score indicated a significant rise in hospital mortality when score exceeded 23 with a mortality rate of 71% (Chen, Hsu, Chen, Fang, & Huang, 2002). Limited generic severity score was applied for critical patients with AKI who requiring RRT (Demirjian et al., 2011; Uchino et al., 2005) and limited prediction accuracy for critically AKI patients requiring dialysis (Chen et al., 2002).   We included SOFA sore but not APACHE II score at dialysis for adjustment and applied a conservative selection criterion(table 3). Variables with P<0.15 were included in further multivariable logistic regression analysis to identify determinants of clinical benefit (complete plus partial clinical success). Stepwise backward variable selection with a significance level of 0.05 for variable retention was used to develop a predictor model. Collinearity of variables was assessed by examining tolerance and the variance inflation factor (VIF) (Liu, Kuang, Gong, & Hou, 2003; Wu et al., 2019). Collinearity of variables from the final model is acceptable since the VIF was between 1.008 and 1.216 (APACHE II vs sepsis 3, SOFA vs sepsis 3). After consulting our statistician Dr. Fu-Chang Hu (International-Harvard Statistical Consulting Company, Taipei, Taiwan.), it is within acceptable levels.

Reference

Chen, Y. C., Chang, S. C., Pu, C., & Tang, G. J. (2013). The impact of nationwide education program on clinical practice in sepsis care and mortality of severe sepsis: a population-based study in Taiwan. PLoS One, 8(10), e77414. doi:10.1371/journal.pone.0077414

Chen, Y. C., Hsu, H. H., Chen, C. Y., Fang, J. T., & Huang, C. C. (2002). Integration of APACHE II and III scoring systems in extremely high risk patients with acute renal failure treated by dialysis. Ren Fail, 24(3), 285-296. doi:10.1081/jdi-120005362

Dellinger, R. P., Carlet, J. M., Masur, H., Gerlach, H., Calandra, T., Cohen, J., . . . Surviving Sepsis Campaign Management Guidelines, C. (2004). Surviving Sepsis Campaign guidelines for management of severe sepsis and septic shock. Crit Care Med, 32(3), 858-873. doi:10.1097/01.ccm.0000117317.18092.e4

Demirjian, S., Chertow, G. M., Zhang, J. H., O'Connor, T. Z., Vitale, J., Paganini, E. P., . . . Network, V. N. A. R. F. T. (2011). Model to predict mortality in critically ill adults with acute kidney injury. Clin J Am Soc Nephrol, 6(9), 2114-2120. doi:10.2215/CJN.02900311

Liu, R. X., Kuang, J., Gong, Q., & Hou, X. L. (2003). Principal component regression analysis with SPSS. Comput Methods Programs Biomed, 71(2), 141-147. Retrieved from http://www.ncbi.nlm.nih.gov/pubmed/12758135

Rhodes, A., Evans, L. E., Alhazzani, W., Levy, M. M., Antonelli, M., Ferrer, R., . . . Dellinger, R. P. (2017). Surviving Sepsis Campaign: International Guidelines for Management of Sepsis and Septic Shock: 2016. Intensive Care Med, 43(3), 304-377. doi:10.1007/s00134-017-4683-6

Uchino, S., Bellomo, R., Morimatsu, H., Morgera, S., Schetz, M., Tan, I., . . . Ending Supportive Therapy for the Kidney, I. (2005). External validation of severity scoring systems for acute renal failure using a multinational database. Crit Care Med, 33(9), 1961-1967. doi:10.1097/01.ccm.0000172279.66229.07

Wu, C. H., Wu, V. C., Yang, Y. W., Lin, Y. H., Yang, S. Y., Lin, P. C., . . . group, T. (2019). Plasma aldosterone after seated saline infusion test outperforms captopril test at predicting clinical outcomes after adrenalectomy for primary aldosteronism. Am J Hypertens. doi:10.1093/ajh/hpz098

Round 2

Reviewer 3 Report

 You have addressed most of my concerns adequately.